

# Evaluation of two fluorescence immunoassays for the rapid detection of SARS-CoV-2 antigen—new tool to detect infective COVID-19 patients

Lorena Porte[1], Paulette Legarraga[1], Mirentxu Iruretagoyena[1],
Valeska Vollrath[1], Gabriel Pizarro[1], Jose Munita[2,3,4], Rafael Araos[2,3,4]
and Thomas Weitzel[1,3]

[1] Laboratorio Clínico, Clínica Alemana, Universidad del Desarrollo, Santiago, Chile
[2] Servicio de Infectología, Clínica Alemana de Santiago, Facultad de Medicina Clínica Alemana, Universidad del Desarrollo, Santiago, Chile
[3] Instituto de Ciencias e Innovación en Medicina (ICIM), Facultad de Medicina Clínica Alemana, Universidad del Desarrollo, Santiago, Chile
[4] Millennium Initiative for Collaborative Research on Bacterial Resistance (MICROB-R), Santiago, Chile

Corresponding authors
Lorena Porte,
lporte@alemana.cl
Thomas Weitzel,
thomas.weitzel@gmail.com

## ABSTRACT

**Background:** Real-Time Reverse-Transcription Polymerase Chain Reaction (RT-PCR) is currently the only recommended diagnostic method for SARS-CoV-2. However, rapid immunoassays for SARS-CoV-2 antigen could significantly reduce the COVID-19 burden currently weighing on laboratories around the world.
**Methods:** We evaluated the performance of two rapid fluorescence immunoassays (FIAs), SOFIA SARS Antigen FIA (Quidel Corporation, San Diego, CA, USA) and STANDARD F COVID-19 Ag FIA (SD Biosensor Inc., Gyeonggi-do, Republic of Korea), which use an automated reader. The study used 64 RT-PCR characterized clinical samples (32 positive; 32 negative), which consisted of nasopharyngeal swabs in universal transport medium.
**Results:** Of the 32 positive specimens, all from patients within 5 days of symptom onset, the Quidel and SD Biosensor assays detected 30 (93.8%) and 29 (90.6%) samples, respectively. Among the 27 samples with high viral loads (Ct ≤ 25), the two tests had a sensitivity of 100%. Specificity was 96.9% for both kits.
**Conclusion:** The high performance of the evaluated FIAs indicates a potential use as rapid and PCR-independent tools for COVID-19 diagnosis in early stages of infection. The excellent sensitivity to detect cases with viral loads above ~$10^6$ copies/mL (Ct values ≤ 25), the estimated threshold of contagiousness, suggests that the assays might serve to rapidly identify infective individuals.

## INTRODUCTION

Since the declaration of the COVID-19 pandemic in March 2020, infection with the new SARS-CoV-2 has resulted in over 30 million confirmed cases and almost 1 million deaths worldwide, as of September 2020 (https://COVID19.who.int). Early detection of

cases by highly sensitive and specific real-time reverse-transcription polymerase chain reaction (RT-PCR) is the currently recommended diagnostic strategy (*World Health Organization, 2020a*). However, the high cost of RT-PCR, shortage of reagents, and need for trained personnel have limited the testing capacities of laboratories to provide results in a timely manner (*World Health Organization, 2020b*). In addition, new aspects of SARS-CoV-2 testing include the estimation of infectivity to tailor control measures of known or suspected COVID-19 cases (*Rhee et al., 2020*). Low viral counts detected by RT-PCR might not correspond to infectious virus, especially in asymptomatic individuals or in patients who continue shedding viral RNA after recovering from COVID-19, resulting in false positives and segregation of people who are no longer infectious (*Jefferson et al., 2020*; *Wáng, 2020*; *Kim et al., 2020*). Therefore, alternative diagnostic tools allowing the rapid testing of large numbers of samples and the ability to evaluate infectivity are of high priority (*European Centre for Disease Prevention and Control (ECDC), 2020*). Recent developments include whole-genome sequencing, CRISPR-based technologies, droplet-based digital PCR, RT-LAMP, and immunoassays for the detection of antibodies or antigen (*Giri et al., 2020*). Rapid antigen detection tests (Ag-RDT) using immunochromatographic (ICT) or fluorescence immunoassays (FIAs) have recently become available; many of which are CE-IVD licensed and some have received FDA emergency use authorization (EUA) (www.finddx.org/COVID-19/pipeline). These tests are fast, easy to use and do not require highly trained personnel or sophisticated equipment. As previously suggested FIAs are highly specific and can reach remarkably high sensitivities, if applied in samples from early phases of infection or with high viral loads (*Porte et al., 2020*; *Weitzel et al., 2020*, *Liotti et al., 2020*). For this reason, they could complement RT-PCR testing by becoming a first line tool for rapid detection of infective individuals.

Here we present the performance of two novel FIA automated antigen detection systems in samples from COVID-19 patients presenting within 5 days of symptom onset.

## MATERIALS AND METHODS

Samples derived from patients attending Clínica Alemana in Santiago, Chile, for COVID-19 testing. Specimens consisted of naso-oropharyngeal flocked swabs obtained by trained personnel and placed in universal transport media (UTM-RT® System; Copan Diagnostics, Murrieta, CA, USA). Samples were examined for SARS-CoV-2 RNA by RT-PCR assay (COVID-19 Genesig, Primerdesign Ltd., Chander's Ford, UK). Samples exhibiting exponential amplification curves and cycle thresholds (Ct) values ≤40 were considered positive (*Porte et al., 2020*).

RT-PCR characterized UTM samples were aliquoted and kept at −80° C until analysis by the two FIA kits, "SOFIA SARS Antigen FIA" (Quidel Corporation, San Diego, CA, USA) and "STANDARD F COVID-19 Ag FIA" (SD Biosensor Inc., Gyeonggi-do, Republic of Korea). Both tests detect SARS-CoV-2 nucleocapsid protein by lateral flow immunofluorescence, which is interpreted by automated analysers (Table 1). The kits are CE-IVD labelled; Quidel recently received EUA by the FDA. Manufacturers state that both tests should be performed using nasopharyngeal swabs collected from symptomatic individuals within 5 days of symptom onset. The use of samples stored in certain brands of

**Table 1 Characteristics of two automated SARS-CoV-2 antigen detection assays.**

| Characteristics | Test N°1 | Test N°2 |
|---|---|---|
| Commercial name | SOFIA SARS Antigen FIA | STANDARD F COVID-19 Ag FIA |
| Manufacturer | Quidel Corporation, San Diego, CA, USA | SD Biosensor Inc., Gyeonggi-do, Republic of Korea |
| Certification | CE-IVD, FDA EUA | CE-IVD |
| Primary specimen[a] | NP or N swab | NP swab |
| Transport medium[a,b] | No | Yes |
| Processing time[c] | 15 min | 30 min |
| Readout | Automated reader SOFIA2 | Automated reader F2400 |

Notes:
FIA, fluorescence immune assay; NP, nasopharyngeal; N, nasal; UTM, universal transport medium; CE-IVD, approved CE marking on in vitro diagnostic medical devices; FDA EUA, US Food and Drug Administration Emergency Use Authorization.
[a] According to manufacturers' recommendations.
[b] Quidel indicates that use of viral transport media may result in decreased test sensitivity; SD Biosensor permits use of verified viral transport medium (including Copan UTM™ used in this evaluation).
[c] Includes incubation and read-out.

universal transport media (including Copan UTM) is permitted for the SD Biosensor assay; the Quidel test initially also allowed using UTM, but a recent instruction update discourages the use of such prediluted samples.

For the evaluation, 32 RT-PCR positive UTM samples, all collected within the first 5 days after symptom onset, and 32 negative specimens were selected. All positive samples were from symptomatic patients, 12 negative samples were from asymptomatic patients screened before surgery. Some of the positive ($n = 27$) and negative samples ($n = 19$) had been used in a previous evaluation (*Weitzel et al., 2020*). Assays were performed using the same sample aliquot, following manufacturers´ instructions, by the same laboratory personnel, who were blinded to RT-PCR results. In brief, specimens were mixed with an extraction reagent, dispensed into the cassette's sample well, and read after incubation by an instrument. Testing procedures were performed under a BSL2 cabinet. RT-PCR served as reference method; in case of discordant results, tests were repeated. Demographic and clinical information was retrieved from the national surveillance system and anonymously analysed. Sensitivity, specificity, and accuracy were calculated as recommended by current Clinical and Laboratory Standards Institute guidelines (*CLSI, 2008*); for the Wilson score Confidence Interval (CI) at 95%, we used OpenEpi (version 3.01). Test performance was evaluated for all samples and for those with high viral loads (Ct ≤ 25), as previously described (*World Health Organization, 2020c*). Kits and analysers were provided by manufacturers at reduced costs for evaluation purposes. The study was approved by the Comité Etico Científico, Facultad de Medicina Clínica Alemana, Universidad del Desarrollo in Santiago, Chile, and a waiver of informed consent was granted (Project ID number 931).

## RESULTS

The study included a total of 64 samples, 32 were RT-PCR positive and 32 RT-PCR negative. The median age was 39 years (IQR 36.7–57) and 52% were male. Median days

**Table 2 Demographic characteristics and RT-PCR results of included patients (*n* = 64).**

| Characteristics/RT-PCR results | |
|---|---|
| Gender | |
| Male | 52% |
| Female | 48% |
| Age (years) | |
| Median | 39 |
| IQR | 36.7–57 |
| Symptom onset to sampling (days) | |
| Median | 2 |
| IQR | 1–3 |
| RT-PCR positive samples (n) | 32 |
| Ct values | |
| Median | 17.95 |
| IQR | 16.4–22.4 |
| Value ≤25 | 29 (90.6%) |

Note:
IQR, inter-quartile range; Ct, cycle threshold.

**Table 3 Performance of two automated SARS-CoV-2 antigen detection assays.**

| Antigen test | | RT-PCR | | | Sensitivity | | | | Specificity | | Accuracy |
|---|---|---|---|---|---|---|---|---|---|---|---|
| Assay | Result | Pos. | | Neg. | All | | High VL[1] | | % | CI 95% | % |
| | | All | High VL[1] | | % | CI 95% | % | CI 95% | | | |
| Sofia SARS Ag FIA | Pos. | 30 | 27 | 1 | 93.8 | [79.9–98.3] | 100 | [87.5–100] | 96.9 | [84.3–99.4] | 95.3 |
| | Neg. | 2 | 0 | 31 | | | | | | | |
| Standard F COVID-19 Ag FIA | Pos. | 29 | 27 | 1 | 90.6 | [75.8–96.8] | 100 | [87.5–100] | 96.9 | [84.3–99.4] | 93.8 |
| | Neg. | 3 | 0 | 31 | | | | | | | |

Notes:
VL, viral load; CI 95%, confidence interval 95%; Pos., positive; Neg., negative.
[1] Samples with Ct ≤ 25.

from symptom onset to RT-PCR testing of positive and negative cases were 2 (IQR 1–3) and 1 (IQR 0.75–4), respectively. Ct values had a median of 17.95 (IQR, 16.4–22.4); 29/32 samples (90.6%) had a Ct ≤ 25 (Table 2).

Both assays demonstrated an overall sensitivity >90%, reaching 100% for samples with high viral loads (Table 3). False negative results were observed with the Quidel and SD Biosensor assays in two and three samples, respectively, which had Cts of 30.89–32.57 and were taken on the 4th or 5th day after symptom onset. Specificity was 96.7% for both tests, that is, both kits displayed a single false positive result, from two distinct symptomatic RT-PCR negative cases (Table 3). The two assays were user friendly, included ready-to-use reagents and required little hands-on time. Moreover, analysers were easy-to-use, stored the results, and included options for QR coding, printing, and connection to laboratory information systems.

## DISCUSSION

Antigen-based assays have risen as one of the most practical alternatives, but independent evaluations of their diagnostic performance are scarce and their role within the routine diagnostic workup is yet not defined (*World Health Organization, 2020c*; *Dinnes et al., 2020*). One of the main concerns about antigen-based tests is that these assays per se have a higher detection threshold than RT-PCR, thus, they might miss cases with low viral replication during very early or late infections (*World Health Organization, 2020c*; *Liotti et al., 2020*). However, the results of this and former studies indicate that antigen detection by immunofluorescence, especially when used with an automated reader, has an excellent sensitivity to detect SARS-CoV-2 in samples with estimated viral loads above ~$10^6$ copies/mL (Ct values ≤ 25) (*World Health Organization, 2020c*), which are found in pre-symptomatic (1–3 days before symptom onset) and early symptomatic COVID-19 cases (up to 5–7 days after symptom onset) (*He et al., 2020*; *Lee et al., 2020*; *Zou et al., 2020*). According to recent modelling studies, elevated viral titers are associated to infectivity (*Larremore et al., 2020*). This is in accordance with in vitro experiments, which showed no viral growth from samples with Cts > 24 or taken >8 days after symptom onset (*Bullard et al., 2020*, *Wölfel et al., 2020*). A viral load of $10^6$ copies/mL has therefore been suggested as the limit of infectivity for clinical practice (*Drosten, 2020*). However, until the exact threshold of contagiousness is known, other authors have considered a more conservative approach (1,000 copies/mL) (*Jacot et al., 2020*).

For samples with high viral loads, our comparative study showed that both tests were 100% sensitive. This is in accordance with manufacturers' information and a recent study from Italy of the STANDARD F assay, which showed a high detection rate of 95.2% (20/21) in samples with Cts < 25 (*Liotti et al., 2020*). In our panel of positive samples, false negatives only occurred with Cts > 30, which translates to viral loads <$10^4$ for the used RT-PCR protocol (*Scohy et al., 2020*), although this finding has to be confirmed with a larger number of specimens. The high-performance value coincides with recent studies of a similar FIA with automated reading (BioEasy, Shenzhen, China), which demonstrated sensitivities of 100% for samples with Cts ≤ 25 and of 98% for samples with Cts ≤ 30 (*Diao et al., 2020*). In contrast, immunochromatographic SARS-CoV-2 antigen tests demonstrated lower sensitivity values of 74–85% for samples with Cts ≤ 25 (*Weitzel et al., 2020*; *Mertens et al., 2020*; *Lambert-Niclot et al., 2020*). The evaluation of test specificity was limited by the low number of included RT-PCR-negative samples. In our experience, the rate of false positives using the Biosensor assay in pre-surgical samples of low-risk asymptomatic patients was 1.6% (6/374) (L. Porte and T. Weitzel, 2020, unpublished observations).

Although additional studies with larger numbers of samples are needed, the excellent performance data of FIA Ag-RDTs suggest their potential use in the following scenarios, when RT-PCR is unavailable or impractical: (1) closed or semi-closed remote communities such as cruise ships or military camps (*World Health Organization, 2020c*), (2) High-risk congregate facilities including schools, care-homes, dormitories, etc., when testing daily or every other day could reduce secondary infections by 100% or

90%, respectively (*Paltiel, Zheng & Walensky, 2020*), and (3) screening of asymptomatic attendees at potential superspreader events, like conferences, weddings, and sports or cultural events. In the future, due to their high sensitivity to detect infective patients, FIA Ag-RDTs might also play an important role within "test-out" strategies, that is, the early release of suspected cases from self-isolation or shortening quarantine for proven cases.

### Funding
The authors received no funding for this work.

### Competing Interests
The authors declare that they have no competing interests.

### Author Contributions
- Lorena Porte conceived and designed the experiments, performed the experiments, analyzed the data, prepared figures and/or tables, authored or reviewed drafts of the paper, and approved the final draft.
- Paulette Legarraga conceived and designed the experiments, performed the experiments, analyzed the data, authored or reviewed drafts of the paper, and approved the final draft.
- Mirentxu Iruretagoyena conceived and designed the experiments, analyzed the data, authored or reviewed drafts of the paper, and approved the final draft.
- Valeska Vollrath conceived and designed the experiments, analyzed the data, authored or reviewed drafts of the paper, and approved the final draft.
- Gabriel Pizarro conceived and designed the experiments, performed the experiments, prepared figures and/or tables, and approved the final draft.
- Jose Munita conceived and designed the experiments, analyzed the data, authored or reviewed drafts of the paper, and approved the final draft.
- Rafael Araos conceived and designed the experiments, analyzed the data, authored or reviewed drafts of the paper, and approved the final draft.
- Thomas Weitzel conceived and designed the experiments, analyzed the data, prepared figures and/or tables, authored or reviewed drafts of the paper, and approved the final draft.

### Ethics
The following information was supplied relating to ethical approvals (i.e., approving body and any reference numbers):

The study was approved by the institutional review board (Comité Etico Científico, Facultad de Medicina Clínica Alemana, Universidad del Desarrollo, Santiago, Chile) and a waiver of informed consent was granted (Project ID number 931).

### Data Availability
Raw data are available as a Supplemental File.

## Supplemental Information

Supplemental information for this article can be found online at http://dx.doi.org/10.7717/peerj.10801#supplemental-information.

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
