# Peer review of "Evaluation of two fluorescence immunoassays for the rapid detection of SARS-CoV-2 antigen—new tool to detect infective COVID-19 patients"

_PeerJ, doi:10.7717/peerj.10801_

## Round 0.1 · original submission · Minor Revisions

Please take into consideration the reviewer’s comments and provide back a point-by-point rebuttal letter addressing those concerns.

Reviewer 1 ·

Basic reporting

Dear Authors,

The word "immune assays", line 41, should be changed to immunoassays like shown on the title of your manuscript.

You could improve the sentence, lines 39 and 40, "However, PCR-independent techniques are urgently needed" by writing "However, rapid immunoassays for SARS-CoV-2 antigen could significantly reduced the COVID-19 testing burden currently weighing on laboratories around the world." This improvement can help an international audience clearly understand the message that you are trying to convey.

This is a minor change, but the sentence, lines 57, 58, and 59, should make clear that the COVID-19 disease did not become a pandemic until 2020. The SARS-CoV-2 virus was first detected in 2019.

Since you are evaluating two commercially available assays, you should add or compare your findings with the performance of each assay as reported by the companies (https://bestbion.com/wp-content/uploads/2020/05/bestbiondx-standardf-covid19-corona-ag.pdf and https://www.quidel.com/immunoassays/rapid-sars-tests/sofia-sars-antigen-fia). Your data appears to match their reported performances.

You considered cycles thresholds (Ct) values ≤40 positive. However, according to Sanguinetti et. al, "considering the current epidemiological scenario, a non-negligible proportion of symptomatic or, most commonly, asymptomatic patients, whose nasopharynx swab samples display Ct values of ≥25–<35 or ≥35, might be negative with STANDARD F COVID-19 Ag FIA (or similar) assays. This scenario would also encompass “new” patients who begin their SARS-CoV-2 infection course with a low viral load (resulting in Ct values of ≥35)." You should cite their paper (https://www.ncbi.nlm.nih.gov/pmc/articles/PMC7510559/).

You could address, in your introduction, the issue of PCR false positives that have been reported in many countries ("reinfections"). This could add to the importance of using rapid SARS-CoV-2 antigen tests (not as a replacement of PCR tests, but complementary). You can find some references here about this big issue:

https://www.cdc.go.kr/board/board.es?mid=a30402000000&bid=0030&act=view&list_no=367267&nPage=1

https://www.ncbi.nlm.nih.gov/pmc/articles/PMC7276353/#r2

https://ophrp.org/journal/view.php?doi=10.24171/j.phrp.2020.11.3.02

https://www.cidrap.umn.edu/news-perspective/2020/05/wha-passes-pandemic-probe-resolution-korea-clarifies-reinfection-reports

https://www.frontiersin.org/articles/10.3389/fcimb.2020.00445/full

https://www.cebm.net/covid-19/infectious-positive-pcr-test-result-covid-19/

Experimental design

Dear Authors,

Your experimental design could be improved by conducting viral cell culture tests along with RT-PCR. However, maybe you do not have access to a biosafety level 3 (BSL 3) laboratory to conduct viral cell culture experiments.

Validity of the findings

Dear Authors,

I think you should have cited previously published research (https://www.ncbi.nlm.nih.gov/pmc/articles/PMC7510559/) that used the same rapid assay(s) and addressed the same issue(s) as your manuscript did.

I think your results should have been validated using viral cell culture. Please use the paragraph bellow as a reference to improve your experimental design in the future.

"Conclusion Prospective routine testing of reference and culture specimens are necessary for each country involved in the pandemic to establish the usefulness and reliability of PCR for Covid-19 and its relation to patients factors. Infectivity is related to the date of onset of symptoms and cycle threshold level. A binary Yes/No approach to the interpretation RT-PCR unvalidated against viral culture will result in false positives with segregation of large numbers of people who are no longer infectious and hence not a threat to public health."

https://www.medrxiv.org/content/10.1101/2020.08.04.20167932v3

Additional comments

Dear Authors,

Your manuscript further validates the importance of using rapid antigen tests for COVID-19 as an important complementary diagnostic method. Rapid SARS-CoV-2 antigen tests could help safely reopen businesses, schools, and sporting events (https://www.nj.com/rutgersfootball/2020/10/big-ten-relies-on-daily-rapid-covid-19-antigen-testing-how-it-works-who-gets-tested-what-positive-tests-mean.html). Your manuscript is relevant to the current COVID-19 global pandemic and is in line with other publications related to COVID-19. This reviewer is conscious of the tremendous amount of work and effort that went into the manuscript. More researchers, like you, should focus on evaluating commercially available rapid antigen tests to help mitigate the global impact that the COVID-19 pandemic is having around the world by knowing the performance of those tests.

Reviewer 2 ·

Basic reporting

This work evaluated the performance of two commercially available antigen based point of care systems that can be potentially used in diagnosis of COVID-19. Both systems are reported to have excellent sensitivity and specificity. The reported work is important but needs significant improvement before it can be considered for publication.

Experimental design

1) The number of samples tested is low (only 64). If possible, I strongly recommend authors to include more samples so that the sensitivity and and specificity data reported will be more reliable.

2) It would be nice to include the details on calculation of specificity and sensitivity in the experimental section.

3) I suggest the information provided in line 113-117 along with more information provided in supporting information to put in a form of table.

Validity of the findings

4) In a recent study, Liotti et al has reported the performance of STANDARD F COVID-19 Ag FIA (SD Biosensor Inc.) with more than 300 nasopharynx samples (DOI:https://doi.org/10.1016/j.cmi.2020.09.030). On the basis of this work, the authors should highlight the importance of their study; which I think is the comparison of analytical performance of two commercially available (and two country of origin) point of care systems.

5) The introduction section is poorly written. I recommend authors to briefly mention the analytical performance of other diagnostics methods (possibly in a separate paragraph) such as CRISPR, ELISA and antibody based methods. A nice summary on the performance of these methods is nicely reported in a recent review [Anal Bioanal Chem. 2020 Sep 18 : 1–14.doi: 10.1007/s00216-020-02889-x ]

6) Line 128-135 can be moved to introduction section.

7) I recommend authors to include the simple working principle or schematics of the (may be in form of figure) assays method somewhere in the introduction or discussion section.

Additional comments

6) Please replace Covid-19 with COVID-19 in all places.
7) In line 41: replace immune assays with immunoassays

---

## Round 0.2 · accepted · Accept

Thanks for addressing the minor revisions requested. Now your manuscript is accepted in PeerJ.

Reviewer 1 ·

Basic reporting

Dear Authors,

Thank you for making the changes that were suggested to improve your manuscript.

Experimental design

No comment

Validity of the findings

Dear Authors,

Thank you for citing the suggested previously published research that used the same rapid assay(s) and addressed the same issue(s) as your manuscript does.

Additional comments

Dear Authors,

Thank you for taking the time and putting the effort to make the suggested changes to your manuscript. Rapid testing is a big issue right now that should be addressed by more researchers like you.

Reviewer 2 ·

Basic reporting

In the revised version, authors have have made/addressed the suggested changes.

Experimental design

In the revised version, authors have have made/addressed the suggested changes.

Validity of the findings

In the revised version, authors have have made/addressed the suggested changes.

Additional comments

Please check for English or any typos (if any) very carefully.